# On the estimation of inverse-probability-of-censoring weights for the evaluation of survival prediction error

**Thomas Prince**[1]*, **Andrea Bommert**[2], **Jörg Rahnenführer**[2], **Matthias Schmid**[1]

**1** Institute for Medical Biometry, Informatics and Epidemiology, Medical Faculty, University of Bonn, Bonn, Germany, **2** Department of Statistics, TU Dortmund University, Dortmund, Germany

* prince@imbie.uni-bonn.de

**Data Availability Statement:** The data underlying the results presented in the simulation study can be generated using the files available from https://github.com/PrinceTh0/IPCW_Brier_estimation.

## Abstract

Inverse probability weighting (IPW) is a popular method for making inferences regarding unobserved or unobservable data of a target population based on observed data. This paper considers IPW applied to right-censored time-to-event data. We investigate the behavior of the inverse-probability-of-censoring weighted (IPCW) Brier score, which is frequently used to assess the predictive performance of time-to-event models. A key requirement of the IPCW Brier score is the estimation of the censoring distribution, which is needed to compute the weights. The established paradigm of splitting a dataset into a training and a test set for model fitting and evaluation raises the question which of these datasets to use in order to fit the censoring model. There seems to be considerable disagreement between authors with regards to this issue, and no standard has been established so far. To shed light on this important question, we conducted a comprehensive experimental study exploring various data scenarios and estimation schemes. We found that it is generally of little importance which dataset is used to model the censoring distribution. However, in some circumstances, such as in the case of a covariate-dependent censoring process, a small sample size, or when dealing with noisy data, it may be advisable to use the test set instead of the training set to model the censoring distribution. A detailed set of practical recommendations concludes our paper.

## Introduction

Individualized risk prediction has become a major topic in medical research. For example, clinicians may use the information obtained from risk prediction models to assess the probability of success or failure of a particular intervention, such as that of checkpoint inhibitor immunotherapy in the treatment of advanced melanoma or pharmacotherapy in the treatment of depression [1, 2] This may result in improved clinical outcomes for individual patients and alleviate the overall burden on healthcare systems, by, e.g., reducing the risk of 30-day hospital readmission [3].

In recent years, multivariable prediction models derived from statistical and machine learning (ML) techniques have gained huge popularity [4–7] Clearly, such models are only useful if

The third-party data underlying the illustrative data example cannot be shared publicly for reasons of confidentiality, as per the regulations of the National Cancer Institute: Surveillance, Epidemiology, and End Results Program. These data are available for researchers who meet the criteria for access to confidential data from https://seer.cancer.gov. Requests for data can be placed via https://seerdataaccess.cancer.gov/seer-data-access.

**Funding:** The author(s) received no specific funding for this work;.

**Competing interests:** The authors have declared that no competing interests exist.

their output closely matches the outcome to be predicted. *Discrimination* and *calibration* are two key metrics used to assess the performance of a diagnostic or prognostic model (e.g., [8, 9]). Discrimination, also known as refinement, measures the ability of a model to discern between individuals that are in fact different according to some clinical outcome or state (e.g., diseased vs. healthy). It is therefore a particularly important metric for diagnostic models and tests, where the goal is to keep the number of false negatives and false positives to a minimum. Calibration on the other hand measures the degree to which a model's predicted probability of a particular outcome matches the observed proportion of said outcome. Both discrimination and calibration are important performance metrics for predictive models (e.g., [10]). In order to assess and optimize the performance of predictive models, various other measures have been proposed, such as $R^2$ coefficients of explained variation, reclassification measures and measures derived from decision curve analysis [see, e.g., 4, 8, 11].

One of the most popular evaluation measures for assessing the performance of predictive models is the *Brier score*, which measures the mean squared distance between the observed frequency of a multicategorical outcome and the respective predicted probabilities, typically in a test sample of size $n_{\text{test}}$ [12]. Importantly, it can be shown that the Brier score combines both discrimination and calibration [13–15], and may also be decomposed into a model-dependent mean-squared-error component and a model-independent variance term [16]. In this paper, we consider the *Brier score for time-to-event models*, which refers to the evaluation of a time-dependent binary outcome measuring the occurrence of a specific target event (e.g., death/survival or response/no response to treatment, [17]). In this setting, the Brier score is equal to the time-dependent squared difference between the binary outcome status and the predicted survival probability, and may be calculated at any time point during a longitudinal study, provided that at least one individual has not yet experienced the event at that particular time. Consequently, the Brier score allows for a continuous evaluation of predicted survival probabilities over the entire study period.

A major challenge in time-to-event modeling arises from the presence of (right-)censored data. Right censoring produces data that is partly missing for a subset of individuals, since the event status is known for these individuals, but only up to the time of censoring. Calculating the Brier score is thus no longer possible for all individuals at all points in time. A popular adaptation of the Brier score designed to address this issue employs the *inverse-probability-of-censoring weighting* (IPCW) technique [18–20]. IPCW is a method designed to mimic a non-censored population based on a censored sample. It is equivalent to artificially inflating the sample of uncensored individuals at each time point $t$ by re-weighting these individuals according to the inverse probability of experiencing a censoring event after $t$, thus creating a pseudo-population of uncensored individuals. Based on the intuitive notion that the probability of an individual being censored increases over time, the rationale is that particular emphasis should be placed on those rare individuals who "survive" the censoring event the longest. Gerds & Schumacher [20] demonstrated that the inverse-probability-of-censoring weighted Brier score is consistent ($n_{\text{test}} \to \infty$) and robust against model misspecification, making it a method of choice for many researchers and analysts.

An important challenge when using the IPCW Brier score is the necessity of having to estimate the censoring distribution, which is used in the IPCW weighting scheme. Importantly, for the IPCW Brier score to be consistent, the censoring distribution must be correctly specified [20]. The question of how to correctly estimate the censoring distribution is far from trivial and continues to be the subject of debate (see, for example, Kvamme & Borgan, [21]). In principle, the problem of estimating the censoring distribution may be tackled in the same way as the estimation of the event time distribution. This is because the estimation of the censoring distribution is simply another time-to-event estimation problem, where the censoring event

effectively becomes the event of interest. Oftentimes a Kaplan-Meier estimator is used, whereby the event status indicator is simply reversed. This method is still widely used due to its simplicity, however, it typically disregards predictors affecting the censoring process, operating under the one-size-fits-all assumption that the censoring distribution is identical for all individuals. As argued by Gerds & Schumacher [20] and Mogensen et al. [22], only estimating the marginal censoring distribution can lead to biased estimates of the IPCW Brier score. As a consequence, more sophisticated time-to-event models may be used to estimate the censoring distribution, such as the Cox proportional hazards model [23], penalized regression (e.g., the Lasso, [24–26]) and machine learning (ML) techniques (e.g., random survival forests, [27], and extreme gradient boosting, [28]). However, this issue lies beyond the scope of the present paper.

Here, we address another key issue in the IPCW-based evaluation of time-to-event models, namely that of which data to use in order to estimate the censoring distribution. Interestingly, this question, which affects virtually all evaluation measures involving IPCW, has thus far received little attention. It arises from the fact that, when fitting and evaluating a new time-to-event prediction model or when performing a benchmark study of prediction models, convention calls for using independent sets of training and test data, possibly as part of a cross-validation scheme (which might also include additional validation data for hyperparameter tuning, see [29, 30]). The training set is used to develop the model and the test set to evaluate its predictions, using, for instance, an IPCW-based evaluation measure like the Brier score. In this setting, an obvious question is whether one should use the training data or the test data to estimate the censoring distribution needed for computing the IPCW Brier score. Several papers exploring and applying the IPCW Brier score, such as the ones by Gerds & Schumacher [20] and Collins et al. [31], use the test set to fit the censoring distribution, but do not provide an explicit rationale for doing so. Of course, one could reasonably argue that the censoring distribution is a key component in the computation of the IPCW Brier score on the test set, and that it must therefore reflect the characteristics of the test set as closely as possible. Consequently, the censoring distribution (and thus also the IPC weights) should be estimated using the test set. However, as pointed out by Kvamme & Borgan [21], fitting the censoring distribution on the test set carries with it the risk of overfitting, especially when employing ML models for fitting the censoring distribution and when the test set is of a smaller size than the training set. Specifically, it contradicts the well-established paradigm that the test data should not be used to perform any model building steps. In light of this, fitting the censoring distribution to the training data may be more appropriate for assessing model performance in some circumstances. As a consequence, some authors have used the training data to estimate the IPC weights (e.g., [21, 32]). Other authors have used all of the data (training and test combined) to fit the censoring model (e.g., [22, 33]). As noted by Kvamme & Borgan [21], a systematic assessment of which approach is more appropriate and leads to better performance estimates is still lacking.

The aim of this paper is to shed light on this important question and to provide "best practice" recommendations on which data to use for estimating the weights for the IPCW Brier score. We begin with notation and a formal definition of the IPCW Brier score (see Section "Notation and Definitions"). Afterwards, we present the results of a comprehensive simulation study analyzing the behavior of the IPCW Brier score with respect to changes in the estimation of the censoring distribution and variations in the sample size and the censoring rate (see Section "Simulation study"). In all of the simulation scenarios we investigate the effect of fitting the censoring model on either the test or the training set or both, comparing the values of the IPCW Brier score to their theoretical counterparts obtained from uncensored data. Using real-world observational data from the Surveillance, Epidemiology, and End Results (SEER)

program [34], we further analyze the behavior of several popular statistical and ML techniques in the estimation of the censoring distribution (see Section "Illustration: SEER breast cancer data"). In particular, we demonstrate that the proper tuning of ML-based censoring models has a major effect on the resulting Brier score values. The main findings of the paper are summarized in the discussion.

## Notation and definitions

### Time-to-event analysis with right-censored data

Consider a test sample of $n_{\text{test}}$ independent and identically distributed individuals, and let $T_i$ and $C_i$ denote the true event and censoring times, respectively, of individual $i \in \{1, \ldots, n_{\text{test}}\}$. Here we consider *right-censored* time-to-event data, meaning that an individual is considered to be censored with respect to the event of interest if their censoring time $C_i$ is less than their true survival time $T_i$. Consequently, the observed event time is defined by $T_i^* = \min\{T_i, C_i\}$. In this paper, we only consider the case of a single target event without competing events. Let $\delta_i$ denote the status indicator variable, whereby $\delta_i = 1$ if the individual is observed to experience the event, otherwise $\delta_i = 0$. We also define a $p$-dimensional vector of time-independent predictors $Z_i = (Z_{i1}, \ldots, Z_{ip})^\top$, which are assumed to influence the event-free survival time of individual $i$. The marginal probability of event-free survival is defined as $S(t) = P(T_i > t)$, whereas the individual-specific probability of event-free survival is defined conditionally on the predictors, i.e., $S_i(t) = P(T_i > t | Z_i = z_i)$, where $t$ corresponds to some point in time. Here we assume that the event and censoring times are conditionally independent, i.e., that the process responsible for generating the target event and that responsible for the occurrence of right censoring are independent of each other given the predictors [20]. Both processes may therefore be modeled as separate probability distributions, where we additionally assume that the two models do not share any common parameters. In this framework, the *censoring survival function* quantifies the time-dependent probability of "surviving" a censoring event: analogously to the definition of $S(t)$, the marginal probability of censoring-free survival is defined as $G(t) = P(C_i > t)$, whereas the individual-specific censoring survival function is defined as $G_i(t) = P(C_i > t | Z_i = z_i)$. In the following, the index $i$ may be omitted in some instances for the sake of simplicity. While estimating the survival function $S(t)$ is naturally of primary concern, especially in predictive modeling, estimation of the censoring distribution is typically of little practical interest. However, as mentioned above, certain methods such as IPC weighting require the estimation of the censoring survival function $G(t)$.

### Inverse-probability-of-censoring weighting

To address the issue of right censoring, we may assign weights to individuals depending on their respective probabilities of "surviving" the censoring event. This weighting method is known as IPC weighting [18, 19]. The intuition behind IPCW is that censored individuals cannot directly contribute to the analysis since the time of the event is unobservable for these individuals. However, it would be unwise to completely ignore censored individuals, since the probability of losing individuals to follow-up increases over time. Subjects with large event times $T_i$ are therefore more likely to be censored than those with smaller event times. Hence, simply excluding censored individuals would lead to a biased sample in which individuals with smaller event times are over-represented, and would thus lead to an underestimation of the survival probabilities. IPCW prevents individuals who are censored at or prior to a particular time point $t$ from contributing to the analysis directly by assigning them a weight of zero, however, the remaining subjects are re-weighted accordingly to account for the excluded censored individuals. Concretely, IPCW requires the estimation of $G(t)$ (or, if a set of predictors is

considered, $G_i(t)$), as the weights assigned to each individual depend on the censoring process. More specifically, for each $t$ the IPC weights $\omega_i$, $i = 1, \ldots, n_{\text{test}}$, assigned to each individual are defined as

$$\omega_i(t) = \frac{\mathbb{I}\{T_i^* \leq t, \delta_i = 1\}}{G_i(T_i^* -)} + \frac{\mathbb{I}\{T_i^* > t\}}{G_i(t)}, \tag{1}$$

where $\mathbb{I}\{\cdot\}$ is the indicator function, and where $T_i^* -$ refers to a time point that is infinitesimally smaller than $T_i^*$. By definition, the IPC weights do not have fixed values but rather vary over time and are thus defined relative to a particular time point $t$. Since the value of $G(t)$ decreases with increasing $t$, individuals with larger observed event times are assigned larger weights. This type of weighting scheme has been adapted for a wide range of methods, including ML techniques [35] as well as the estimation of performance metrics like the Brier score and the concordance statistic for time-to-event data ([36], see also the Discussion for a brief overview).

## The Brier score

The Brier score is a scoring metric frequently used for evaluating the accuracy of predictions in classification, time-to-event analysis and other settings. In the case of survival data with a single target event, the Brier score is equivalent to the *mean squared error* (*MSE*), which is defined by the mean squared difference of the true and the estimated survival probabilities, $S_i(t)$ and $\hat{S}_i(t)$, evaluated at time $t$:

$$\text{MSE}(t) = \frac{1}{n_{\text{test}}} \sum_{i=1}^{n_{\text{test}}} \left( S_i(t) - \hat{S}_i(t) \right)^2. \tag{2}$$

While the estimated survival probabilities $\hat{S}_i$ may be derived from the data, the true survival probabilities $S_i(t)$ are generally unknown and must therefore be approximated through the use of step functions which are equal to one prior to the event times and zero afterwards. The empirical Brier score for uncensored data, which we will refer to as the *uncensored Brier score* in this paper, is therefore defined as

$$\text{BS}(t) = \frac{1}{n_{\text{test}}} \sum_{i=1}^{n_{\text{test}}} \left( \mathbb{I}\{T_i^* > t\} - \hat{S}_i(t) \right)^2. \tag{3}$$

Replacing the true survival function with an approximation inevitably introduces some amount of bias. However, it can be shown that this bias depends on $S_i(t)$ but not $\hat{S}_i(t)$ [21, 37]. More specifically, the expectation of the uncensored Brier score is equal to

$$\mathbb{E}(\text{BS}(t)) = \text{MSE}(t) + \frac{1}{n_{\text{test}}} \sum_{i=1}^{n_{\text{test}}} S_i(t)(1 - S_i(t)), \tag{4}$$

where the second term in (4) is the *irreducible loss*. The proof of (4), as well as a decomposition of the uncensored Brier score into calibration and refinement (i.e., discrimination) terms, is given in S1 File.

Obviously, the computation of the uncensored Brier score in (3) is only feasible in datasets where all of the event times are observed, i.e., where $T_i = T_i^* \ \forall i = 1, \ldots, n_{\text{test}}$. When dealing with right-censored data, we may use inverse-probability-of-censoring weighting to

approximate the Brier score for uncensored data. The *IPCW Brier score* is defined as

$$\mathrm{BS}_{IPCW}(t) = \frac{1}{\tilde{n}(t)} \sum_{i=1}^{n_{\text{test}}} \left( \frac{\hat{S}_i(t)^2 \, \mathbb{I}\{T_i^* \leq t, \delta_i = 1\}}{G_i(T_i^*-)} + \frac{(1 - \hat{S}_i(t))^2 \, \mathbb{I}\{T_i^* > t\}}{G_i(t)} \right), \tag{5}$$

where $\tilde{n}(t)$ corresponds to the sum of the IPC weights in (1). Note that when estimating the IPC weights on the test set with a simple method such as the Kaplan-Meier estimator, $\tilde{n}$ is equal to $n$ [21]. It can be shown that the expectation of the IPCW Brier score is equal to the expectation of the Brier score for uncensored data if the functions $G_i(\cdot)$, $i = 1, \ldots, n_{\text{test}}$, are known. When the functions $G_i(t)$ are unknown, which is most often the case, we may replace $G_i(t)$ in Eqs (1) and (5) above with an estimator $\hat{G}_i(t)$. If the censoring model is correctly specified, i.e., if the estimator $\hat{G}_i(t)$ is weakly consistent for $G_i(t)$, the expectation of (5) will converge to (4) for $n_{\text{test}} \to \infty$ [20]. Note that, unlike in other settings with a binary outcome where the Brier score corresponds to a single value, the Brier score in time-to-event analysis consists of a series of scores, as it depends on the time $t$. These scores may be plotted over time to illustrate the change of prediction error with increasing $t$ ("prediction error curve", [22]).

## Training and test data

As stated in the introduction, one of the persisting questions regarding the application of the IPCW Brier score concerns the estimation of the censoring distribution. Here, we aim to determine whether the training set, the test set or the combined dataset is best suited for this purpose. We generally assume that each pair of training and test sets is the result of randomly subsampling from a single "parent" dataset. Additionally, we assume that fitted models of the survival function are obtained on the training sample of size $n_{\text{training}}$. Denoting the estimated survival function by $\hat{S}$ (or alternatively by $\hat{S}_i$, depending on the context), predictions are then computed on an independent test sample of size $n_{\text{test}}$. Models of the censoring survival function are obtained on either the training, test or combined sample, depending on the considered scenario, and predictions for the estimated censoring survival probabilities $\hat{G}$ (or $\hat{G}_i$, depending on the context) are computed on the test set. Further details regarding the model fitting and estimation process are given in Section "Simulation study" and Section "Notations and Definitions". In some instances, we also employ bootstrapped datasets of the training and test data, see also Section "Data preparation and modeling" of the SEER data example. In cases where prior tuning of hyperparameters is necessary, e.g., for the random forest and extreme gradient boosting models, we perform cross-validation and Bayesian optimization on the dataset used for model fitting.

## Simulation study

Given the lack of best-practice recommendations with regards to the estimation of the censoring survival function $G(t)$ for the IPCW Brier score, we conducted a comprehensive simulation study to gain further insight into how to best approach this issue. Concretely, we aimed to (a) investigate whether the training set, the test set or the combined dataset is best suited for fitting the censoring model, and to which degree other parameters such as the sample size or the censoring rate affect the performance estimates, (b) determine the impact of misspecification of the censoring model on the performance of the IPCW Brier score and (c) explore the effect of fitting the censoring survival function with more complex ML estimators, in particular when the data contains many non-informative predictors. In the following, we will present the data generating process used for creating the right-censored time-to-event datasets, the performance metrics employed to assess the performance of the IPCW Brier score and a detailed

step-by-step summary of the simulation pipeline. The corresponding R code is available on Github (https://github.com/PrinceTh0/IPCW_Brier_estimation).

## Generating right-censored time-to-event data

We used the Weibull and the exponential distribution to generate our right-censored datasets. Let $Y_i$ equal the natural logarithm of Weibull-distributed time, i.e., $Y_i = \log(T_i)$, and let $Z_i \in \mathbb{R}^p$ and $\gamma^\top = (\gamma_1, \ldots, \gamma_p)$ represent the matrix of predictors and the vector of corresponding regression coefficients. For the purpose of our study, we represent the link between log time and the predictors by a linear relationship of the form

$$Y_i = \log(T_i) = \mu + \gamma^\top Z_i + \sigma W_i, \tag{6}$$

where $\mu \in \mathbb{R}$ and $\sigma > 0$ correspond to the intercept and scale parameter of the model, respectively. As shown by Klein & Moeschberger [38], $W_i$, which may be viewed as a residual term, follows a standard extreme value distribution with density $f_W(w) = \exp(w - \exp(w))$. In particular, it may be shown that when the natural logarithm of the event time $\log(T_i)$ follows an extreme value distribution, then $T_i$ follows a Weibull distribution with shape parameter $\alpha = 1/\sigma$ and scale parameter $\lambda = \exp(\mu + \gamma^\top Z_i)$. In the special case of the exponential distribution, $\sigma$ is set to 1. By scaling the negative regression coefficients $-\gamma$, we obtain a proportional hazards model with regression coefficients $\beta = -\gamma/\sigma$ and a Weibull hazard function of the form

$$h(t|Z_i) = \alpha \lambda t^{\alpha-1} \exp^{\beta^\top Z_i}, \tag{7}$$

where $\alpha \lambda t^{\alpha-1}$ corresponds to the baseline hazard function $h_0(t)$. We use this kind of model to generate both the event time and the censoring distributions. In the case of the censoring distribution, the random variable $T_i$ is simply replaced by $C_i$ and the values of the intercept and scale parameters $\mu$ and $\sigma$ are readjusted to control the rate of censoring and the coefficient of determination $R^2$, which we defined as $\text{var}(\beta^\top Z)/\text{var}(\log(T))$ (omitting the index $i$ for the sake of simplicity).

## Simulation parameters and settings

Our datasets included a set of $p = 10$ informative predictors. The values of the Weibull regression coefficients were fixed at $\gamma = (-0.5, -0.4, -0.3, -0.2, -0.1, 0.1, 0.2, 0.3, 0.4, 0.5)^\top$. These were identical for the event and censoring distributions. The $p$ predictors in each dataset were drawn from a multivariate normal distribution with zero mean and a compound symmetry covariance structure ($\text{var}(Z_j) = 1, 1 \leq j \leq 10, \text{cov}(Z_j, Z_k) = 0.5, j \neq k, 1 \leq j, k \leq 10$). In some of our scenarios we added a set of $q = 50$ non-informative predictors which neither contributed to the generation of the event times nor of the censoring times (for an overview of the simulation scenarios see Section "Model fitting and prediction" of the simulation study). These were designed to emulate "noisy" data. For the distribution of the event times, we set the value of the scale parameter $\sigma_T$ to approximately 0.58, which resulted in a value for $R^2$ of approximately 0.5 across all of our experiments, while the value of the intercept parameter $\mu_T$ was set to 0. We used two different censoring models, either a marginal model or a "full" Weibull model. In the marginal model, the censoring times $C_i$ were independent of the predictors. In this case, we set the value of the scale parameter $\sigma$ to 1, resulting in an exponential distribution. In the "full" model on the other hand, the values of the predictors influenced the censoring times $C_i$, leading to hazard functions that varied from individual to individual. For these models, we set the value of $\sigma_C$ to approximately 0.58, resulting in $R^2 \approx 0.5$ for the censoring distribution. We set the level of censoring to approximately either 20%, 50% or 80%, depending on the scenario. This was done by offsetting the value of the location parameter $\mu_C$ for the full censoring models

and by adjusting the rate parameter $\lambda_C$ for the marginal (predictor-free) censoring models. The location parameter was set to either 0.800, −0.803, or −0.002 and the rate parameter to either 0.219, 0.864, or 3.013. Since the level of censoring is also marginally affected by the scale parameter $\sigma$, the values of these three parameters were jointly determined in advance using large external datasets of size $10^6$.

## Model fitting and prediction

For each scenario we generated $m = 1000$ time-to-event datasets with $n$ observations each. Each dataset of size $n$ was randomly split into a training and a test dataset. In order to determine whether the relative size of the training set ($n_{\text{training}}$) and of the test set ($n_{\text{test}}$) had any effect, we varied the size of one of these datasets, with values chosen among {25, 50, 100, 250, 500}, while keeping the size of the other fixed at 500. Thus, the total size of each combined dataset (training and test) varied between $n = 525$ and $n = 1000$. Subsequently, a time-to-event model was fit to the training data. We used several popular and well-established methods, including regression-based models such as the Cox proportional hazards model [23] and regularized Cox regression (here, the Lasso, [24–26]), as well as ML methods such as random survival forests [27] and extreme gradient boosting (XGBoost, [28]). With the exception of the Cox proportional hazards model, which does not require prior tuning of hyperparameters, all models were tuned using cross-validation and/or Bayesian optimization, see [39, 40]. Further details regarding the tuning algorithm, priors and the chosen hyperparameters are provided in S2 File. The resulting model was then used to predict the survival function on the test set. The censoring model was fit to either the training set, the test set, or the combined dataset. This was done for both the marginal model, using a Kaplan-Meier estimator, as well as for the full model using either a Cox proportional hazards model, Lasso, random forest or XGBoost. Each of the three resulting models, obtained on either the training, test or combined dataset, was then used to provide a prediction of $\hat{G}(t)$ on the test set. As a result, we obtained an estimate of the survival function $\hat{S}(t)$, as well as three distinct estimates of the censoring survival function ($\hat{G}_{training}(t)$, $\hat{G}_{test}(t)$ and $\hat{G}_{combined}(t)$), corresponding to the censoring survival function fitted to the training, test or to the combined dataset, respectively. In total, we considered five different scenarios in our simulations:

(i). a baseline scenario with an exponentially distributed and predictor-free censoring process, where the event time model was fitted using a Cox regression model and the censoring model using a Kaplan-Meier estimator,

(ii). a Weibull-Cox scenario with Weibull-distributed and predictor-dependent censoring processes, where both the event time and the censoring model were fitted using Cox regression models,

(iii). a misspecification scenario with Weibull-distributed and predictor-dependent censoring processes, where the event time model was fitted using a Cox regression model but the censoring model was "erroneously" fitted using the (marginal) Kaplan-Meier estimator,

(iv). a "low-noise" scenario with Weibull-distributed and predictor-dependent censoring processes, where both the event time and the censoring model were fitted using statistical modeling and ML methods (Cox regression, Lasso, random forest and XGBoost), and

(v). a "high-noise" scenario, which extended the low-noise scenario by 50 additional non-informative predictors.

## Performance metrics

We assessed the effect of the different estimates of $G_i(t)$ on prediction error by comparing the IPCW Brier score (Eq 5) obtained for each estimate of $G_i(t)$ to the expectation of the uncensored Brier score (Eq 4). This was possible due to the fact that in simulation settings the true event times $T_i$, $i = 1, \ldots, n$, and the true survival functions $S_i(t)$ are known for all individuals. In order to standardize the comparison of differently parameterized distributions and to allow for an unbiased comparison of Brier score curves obtained on different datasets, we introduce a weighted and standardized cumulative measure of prediction error aimed at providing a single score for performance comparison. This measure represents the weighted squared difference of the IPCW Brier score obtained on right-censored data and the expectation of the uncensored Brier score obtained on a comparable set of uncensored data. In the following, we will refer to this measure as the *weighted squared error* (WSE), which is formally defined as

$$\text{WSE}(t_j) = \frac{\left(E(\text{BS}(t_j)) - \text{BS}_{IPCW}(t_j)\right)^2 \sum_{i=1}^{n}(S_i(t_{j-1}) - S_i(t_j))}{\sum_{j=1}^{k}\sum_{i=1}^{n}(S_i(t_{j-1}) - S_i(t_j))} \; . \tag{8}$$

In practice, each instance of the squared error $(E(\text{BS}) - \text{BS}_{\text{IPCW}})^2$ is weighted by an estimate of the underlying marginal probability density of the event times. In this particular case, the term "instance" refers to the Brier score or squared error evaluated at any time $t_j$, $j = 1, \ldots, k$. We chose a grid of $10^4$ equidistant time points for the calculation of the Brier scores and the WSE, therefore $k = 10^4$ in all of our simulations. In order to ensure a fair comparison, we used a separate external dataset of size $10^6$, obtained using the same data-generating process as that used for generating the original dataset, in order to (numerically) determine the uncensored Brier score. By summing over all $k$ instances and by taking the square root, we obtain a time-independent score defined by

$$\text{RWSE} = \sqrt{\sum_{j=1}^{k}\text{WSE}(t_j)} \, , \tag{9}$$

which was used as the main performance measure in our simulations. In order to prevent large individual IPC weights from skewing the results, we capped the value of the weights $\omega_i(t)$ at 5 (see [21, 41]).

## Results

The main results for our baseline scenario (i) are presented in Fig 1. As with the following scenarios, we chose to visualize the Monte-Carlo error using violin plots. It can be seen that the RWSE does not systematically differ depending on the dataset used for fitting the censoring distribution. This seems to be the case regardless of the sample size of each dataset ($n$), the relative size of the training and test sets ($n_{\text{training}}$ and $n_{\text{test}}$, respectively), as well as the censoring rate. Unsurprisingly, the estimation accuracy of the IPCW Brier Score improves with an increase in the total sample size $n$ and a lower censoring rate. The reduction of the RWSE is more pronounced following an increase in the size of the training set $n_{\text{training}}$ with a constant $n_{\text{test}}$ (Fig 1b), compared to an increase in the size of the test set $n_{\text{test}}$ with a constant $n_{\text{training}}$ (Fig 1c). This may be due to the fact that a larger training set leads to a better estimate of the survival function $S(t)$, which is likely more important for improving the Brier score estimate than an improved estimate of the censoring survival function $G(t)$. Further numerical results for this baseline scenario may be found in S1 Table.

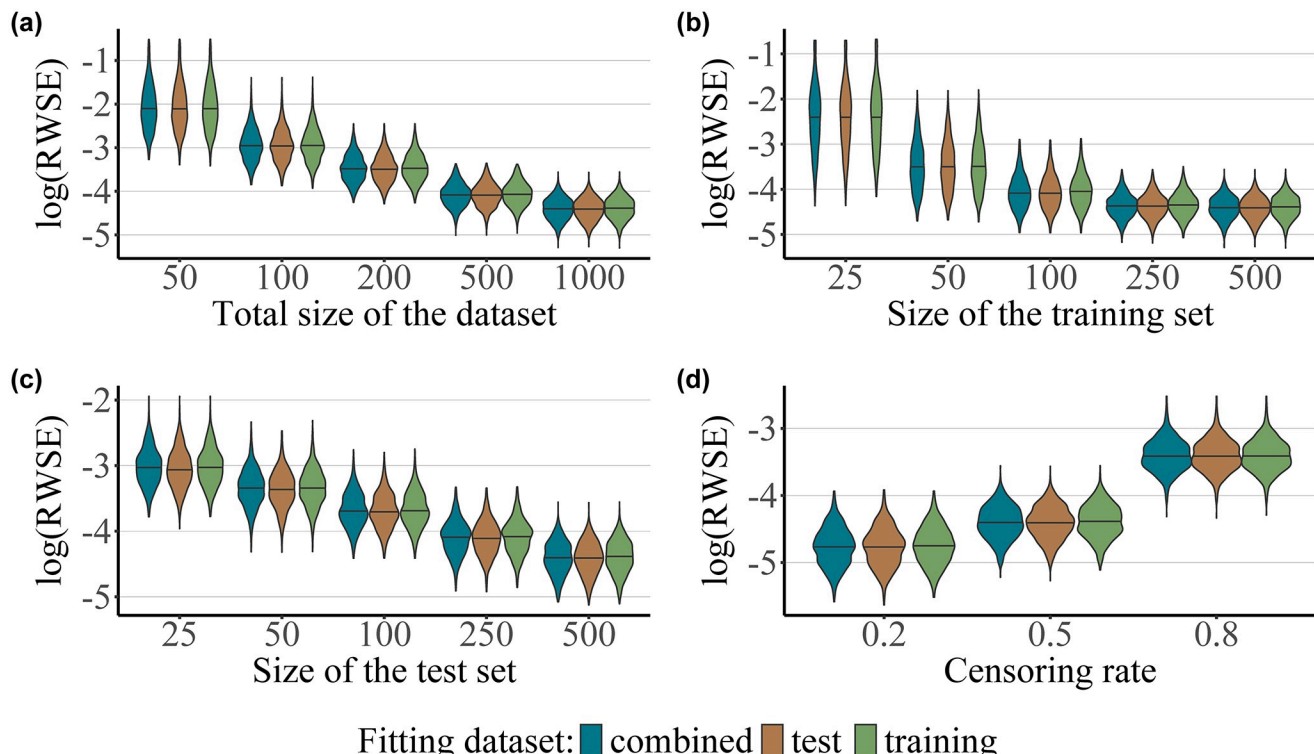

**Fig 1. Baseline scenario with an exponentially distributed and predictor-free censoring process.** The plots present values of the log-transformed RWSE score obtained for 1000 simulation runs as a function of the dataset used for fitting the censoring survival function (test, training or the combined dataset), for different values of $n$, which corresponds to the size of the combined dataset (with $n_{\text{training}} = n_{\text{test}}$, panel a), for different values of $n_{\text{training}}$ and a constant value for $n_{\text{test}} = 500$ (panel b), for different values of $n_{\text{test}}$ and a constant value for $n_{\text{training}} = 500$ (panel c), and for different values of the censoring rate (20%, 50% and 80%, where $n_{\text{training}} = n_{\text{test}} = 500$, panel d). The survival function was fitted using a Cox proportional hazards model, whereas the censoring survival function was fitted using a marginal Kaplan-Meier estimator.

Fig 2 presents the results for the Weibull-Cox scenario (ii) with 10 informative predictors. Similarly to the results obtained for the baseline scenario, the estimation accuracy of the IPCW Brier score is again strongly dependent on the overall size of the dataset, with smaller sample sizes leading to a particularly large deviation from the expected prediction error, regardless of the dataset used for fitting the censoring distribution, see Fig 2. Interestingly, unlike in the baseline scenario, the data used to fit the censoring distribution does seem to affect overall estimation accuracy. Concretely, the estimators where the censoring survival function was fit to the test set slightly outperform those fit to the combined dataset, which themselves outperform those fit to the training set. Additionally, increased rates of censoring once again lead to an increase in estimation accuracy. At low to medium rates of censoring ($\leq 50\%$), the estimators where the censoring distribution was fit to the test set slightly outperform those where the censoring distribution was fit only to the training set. Further numerical results for the Weibull-Cox scenario are presented in S1 Table.

Another important aspect that merits attention is the effect of model misspecification on prediction error. We consider a model to be misspecified if it is structurally incapable of modeling some or most of the complexity contained in the true data generating process. We examined this by comparing the results of the misspecification scenario (iii) with those of the Weibull-Cox scenario (ii). As expected, the correctly specified Cox model in the Weibull-Cox scenario outperforms the Kaplan-Meier estimator in the misspecification scenario, especially

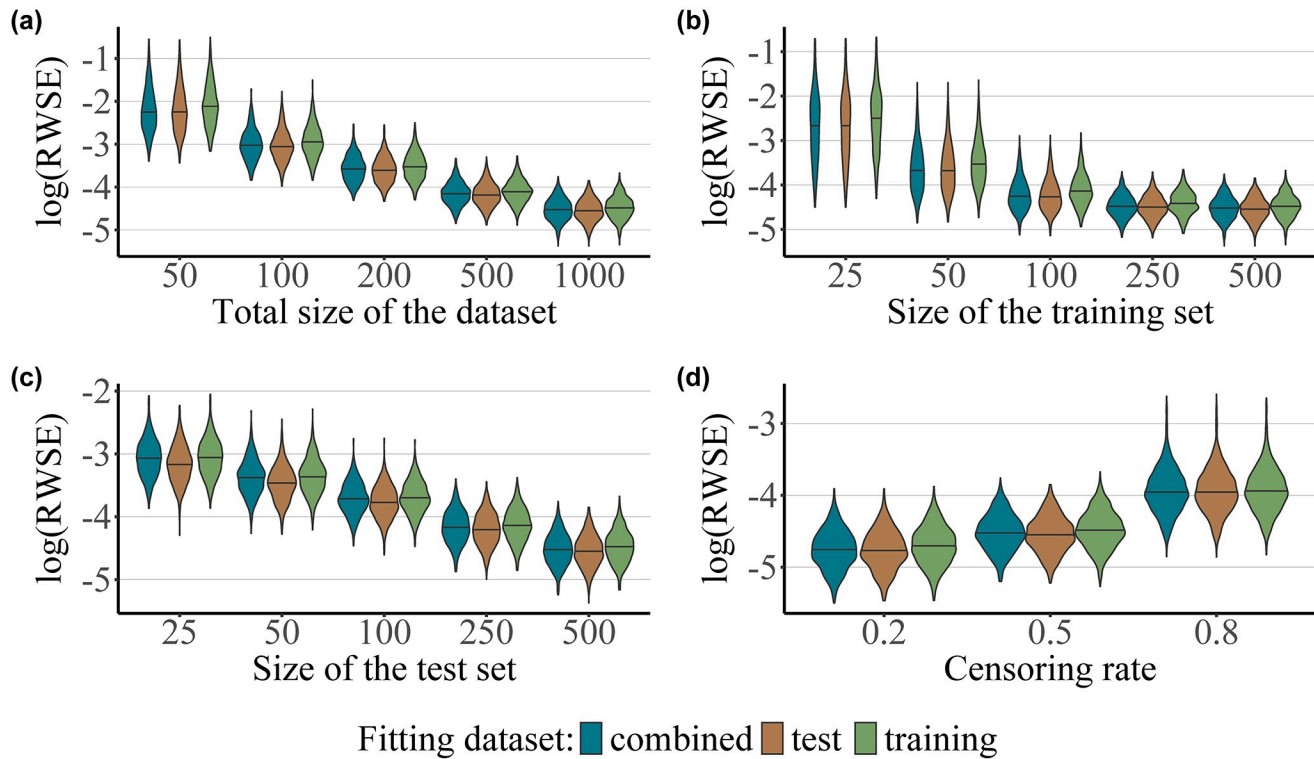

**Fig 2. Weibull-Cox scenario with a Weibull distributed and predictor-dependent censoring process.** The plots present the values of the log-transformed RWSE score obtained for 1000 simulation runs as a function of the dataset used for fitting the censoring survival function (test, training or the combined dataset), for different values of $n$, which corresponds to the size of the combined dataset (with $n_{\text{training}} = n_{\text{test}}$, panel a), for different values of $n_{\text{training}}$ and a constant value for $n_{\text{test}} = 500$ (panel b), for different values of $n_{\text{test}}$ and a constant value for $n_{\text{training}} = 500$ (panel c), and for different values of the censoring rate (20%, 50% and 80%, where $n_{\text{training}} = n_{\text{test}} = 500$, panel d). Both the survival and the censoring survival function were fitted using a Cox proportional hazards model.

for larger datasets (n $\geq$ 100), see Fig 3, stressing the importance of incorporating all of the available covariate information when modeling the censoring process. Importantly, as in the case of a correctly specified model (scenario (ii)), the estimators fit to the test data also slightly outperform those fit to the training set in the misspecification scenario. Further numerical results for the misspecification scenario are presented in S1 Table.

In order to assess the effects of estimates of the censoring distribution in the context of ML and noisy data, we selected four popular methods widely used in the field, namely Cox regression, regularized Cox regression (Lasso), random forest, and extreme gradient boosting (XGBoost). Details on the tuning of the hyperparameters of the Lasso, random forest and XGBoost models are presented in S2 File. Each type of model was applied separately to fit both the survival and the censoring function, on either the training, test or the combined dataset, whereby $n = 1000$, $n_{\text{training}} = 500$, $n_{\text{test}} = 500$, and the censoring rate $\approx 50\%$ for all models. In our low-noise scenario (iv), the underlying survival and censoring functions were both Weibull distributed and included a set of 10 informative predictors each. The results are presented in Fig 4. Similarly to the Weibull-Cox scenario (Fig 2), there is an advantage in using the test set in order to fit the censoring survival function, as opposed to using the training set or the combined dataset when dealing with data with a low signal-to-noise ratio. However, we only found this effect for the Cox proportional hazards and random forest models. Further numerical results for the low-noise scenario are presented in S1 Table.

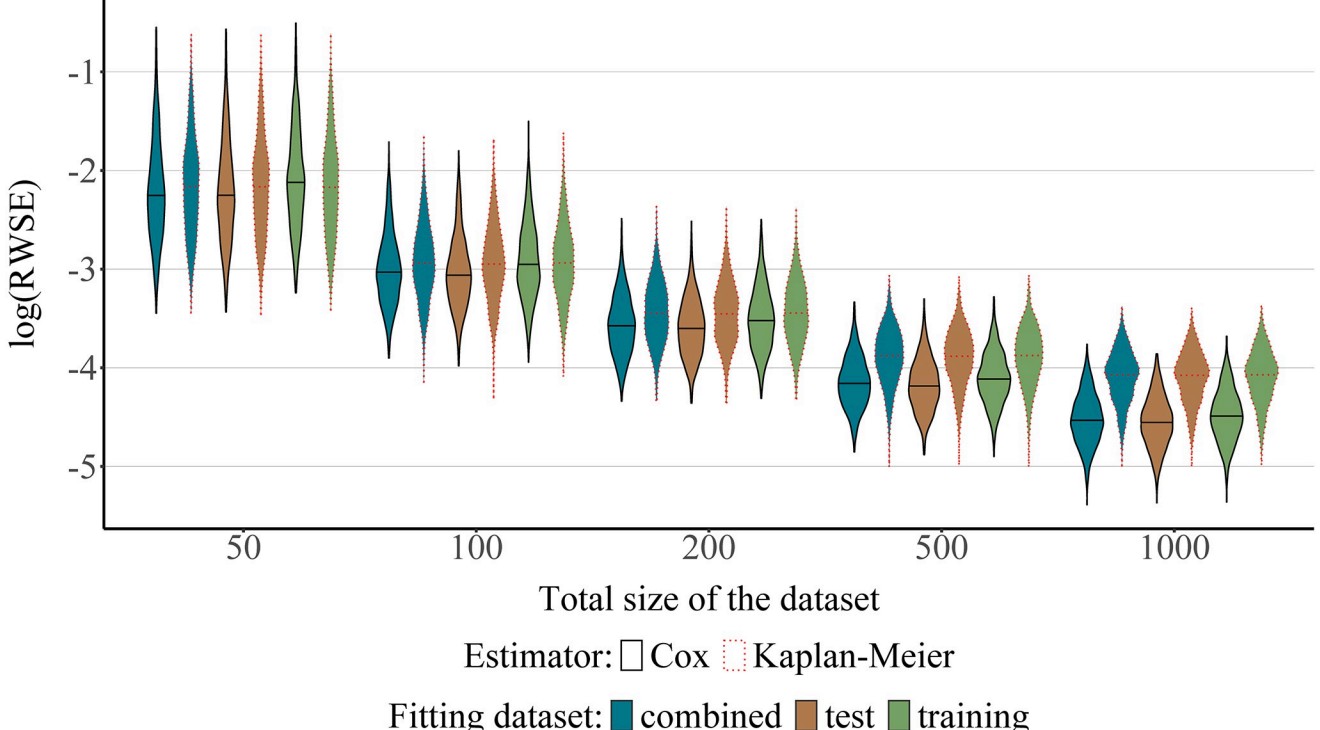

**Fig 3. Weibull-Cox and misspecification scenarios with a Weibull distributed and predictor-dependent censoring process.** The plot presents the values of the log-transformed RWSE score obtained for 1000 simulation runs as a function of the dataset used for fitting the censoring distribution (test, training or the combined dataset), and as a function of the type of model used to fit the censoring function (Cox regression [solid lines, correctly specified] or Kaplan-Meier estimator [dashed lines, misspecified]). The total size of the dataset is given by $n = n_{\text{training}} + n_{\text{test}}$, where $n_{\text{training}} = n_{\text{test}}$.

We subsequently repeated the above analysis for the high-noise scenario (v). Again, as seen in Fig 5, there is an advantage in using the test set in order to fit the censoring survival function, as opposed to only using the training set when dealing with data with a low signal-to-noise ratio. This is particularly evident for Cox proporional hazards (panel a), random forest (panel c) and XGBoost models (panel d), but not for the Lasso (panel b). Further numerical results for the high-noise scenario are presented in S1 Table.

## Illustration: SEER breast cancer data

For a further illustration on a real-world dataset, we analyzed the 2013 breast cancer data from the Surveillance, Epidemiology and End Results (SEER) program of the US National Cancer Institute [34]. The SEER program has been collecting data on cancer incidence, treatment and survival since 1973, covering close to 50% of the US population. We will begin by describing the initial data preparation steps, including variable selection, followed by a description of the model fitting and estimation process and a presentation of our main findings.

### Data preparation and modeling

For our analysis, we selected female breast cancer patients aged between 18 and 75 years at the time of diagnosis who first entered the database between 1998 and 2010. We excluded patients with distant metastases and those without a clear histological confirmation of the diagnosis. Of the 143 variables included in the original dataset, we selected 20 to use in our analyses [42].

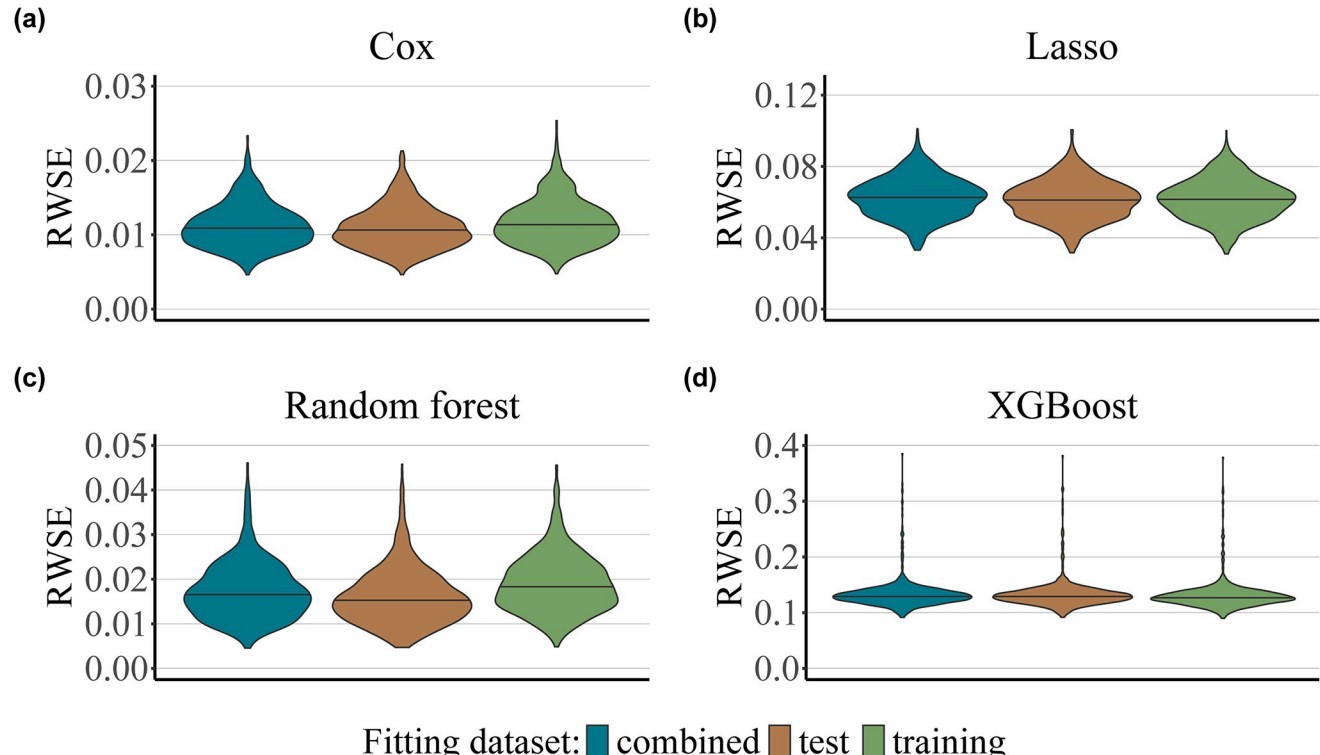

**Fig 4. Low-noise scenario.** The plots present the values of the RWSE score obtained for 1000 simulation runs as a function of the dataset used for fitting the censoring distribution (test, training or the combined dataset), and as a function of the type of model used to fit the survival and the censoring survival functions (Cox proportional hazards, panel a; Lasso, panel b; random forest, panel c; XGBoost, panel d). The total size of the dataset is given by $n = n_{training} + n_{test} = 1000$, where $n_{training} = n_{test} = 500$. The censoring rate was set to approximately 50%.

The continuous and ordinal predictor variables included in our dataset consisted of the age of the patient at diagnosis (years), the size of the tumor (mm), the number of positive and examined lymph nodes, and the number of primaries. Categorical features included cancer stage according to the TNM classification system (12 T-stage and 4 N-stage categories), tumor grade (I—IV), estrogen and progesterone receptor status (positive/negative), primary tumor site (9 categories), surgery of primary site (yes/no), type of radiation therapy and sequence of radiation and/or surgery (7 and 6 categories, respectively), laterality (left/right), ethnicity (white, black, American Indian/Alaska Native, Asian or Pacific Islander, unknown), Spanish origin (nine categories), and marital status at diagnosis (single, married, separated, divorced, widowed). For the TNM T-stage, Hispanic origin and radiation/surgery sequence variables, similar categories with relatively few observations were collapsed into larger classes (see [42]). Individuals with missing observations for either the predictor or the outcome variables were excluded from the analysis. The final analysis dataset included 121, 798 observations, with a median survival time of 63 months (min: 1, max: 156) and a censoring rate of approximately 93.1%.

We randomly split the dataset into a training and a test set of equal size ($n_{training} = n_{test} = 60, 899$, to rule out possible effects linked to different sample sizes) and generated 10 bootstrap datasets each. Using either Cox proportional hazards regression, random forests, or extreme gradient boosting, we fitted a survival model to the bootstrapped training data and a censoring model to either the bootstrapped training, test or combined dataset, analogously to the steps

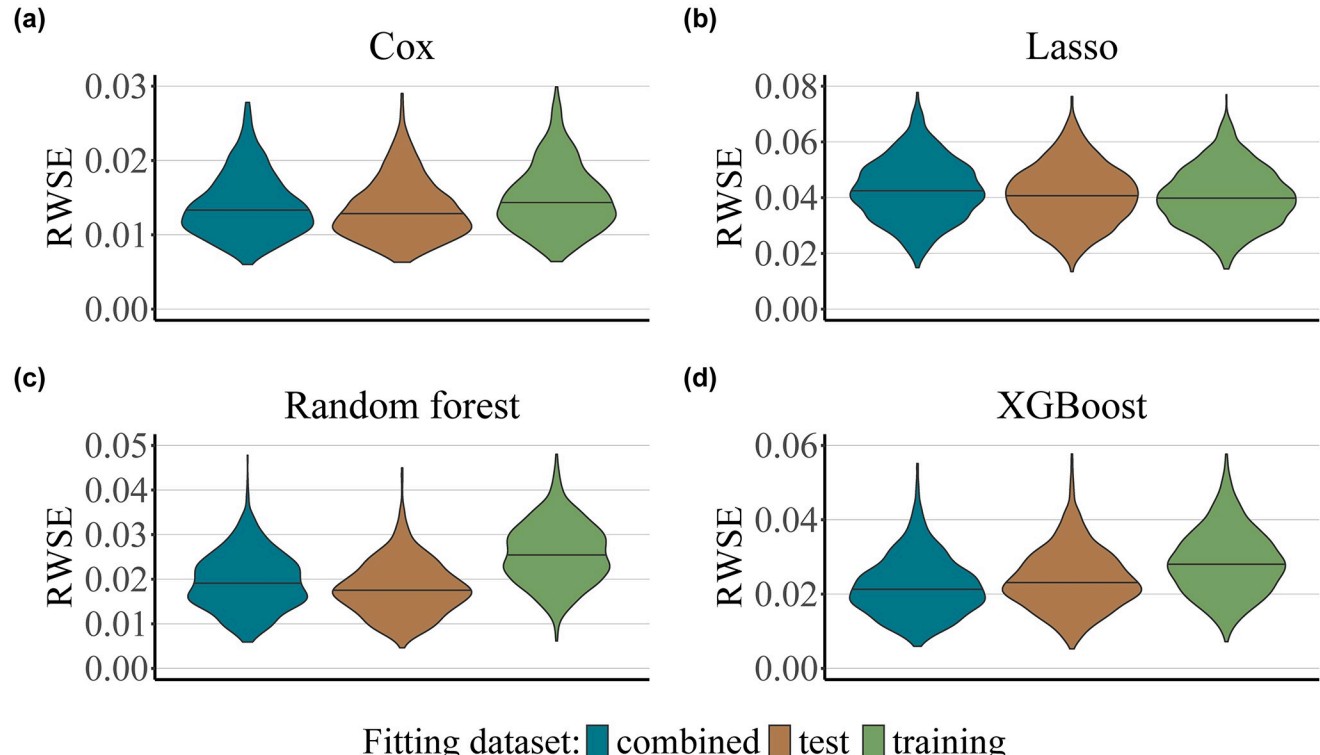

**Fig 5. High-noise scenario.** The plots present the values of the RWSE score obtained for 1000 simulation runs as a function of the dataset used for fitting the censoring distribution (test, training or the combined dataset), and as a function of the type of model used to fit the survival and the censoring survival function (Cox proportional hazards, panel a; Lasso, panel b; random forest, panel c; XGBoost, panel d) on a dataset with a signal-to-noise ratio among the predictors of 1:5 (10 informative and 50 non-informative predictors). The total size of the dataset is given by $n = n_{\text{training}} + n_{\text{test}} = 1000$, where $n_{\text{training}} = n_{\text{test}} = 500$. The censoring rate was set to approximately 50%.

described previously for the simulated survival data. Note that in contrast to the simulation study, we did not consider a penalized Cox model, i.e., Lasso, because of the lack of a natural definition of an untuned model with "default" hyperparameters, so that a comparison of a tuned and untuned model would not have been possible in a natural way. For the tuned models, prior to fitting the models, the main hyperparameters of the random forest and XGBoost algorithms were tuned using a Bayesian optimization algorithm [39, 40] on each dataset. For the untuned models we used the default settings provided in the respective R packages. Further details regarding the tuning algorithm, priors and the chosen hyperparameters are provided in S2 File. Predictions for $\hat{S}(t)$, $\hat{G}_{\text{training}}(t)$, $\hat{G}_{\text{test}}(t)$ and $\hat{G}_{\text{combined}}(t)$ were obtained on the test data. The IPCW Brier score was then computed for each of the censoring models and averaged over the 10 bootstrapped datasets. Further information on the SEER program and the available data can be found on the SEER website (https://seer.cancer.gov). The R scripts with the pre-processing steps and the computation of the IPCW Brier score are available on Github.

## Results

The following results provide an analysis of the predictive performance of three estimation methods—Cox proportional hazards, random forest and XGBoost—as a function of time and of the estimated censoring survival function using the IPCW Brier score. In contrast to the simulation setting, where the true underlying survival probabilities $S_i(t)$ are known and the

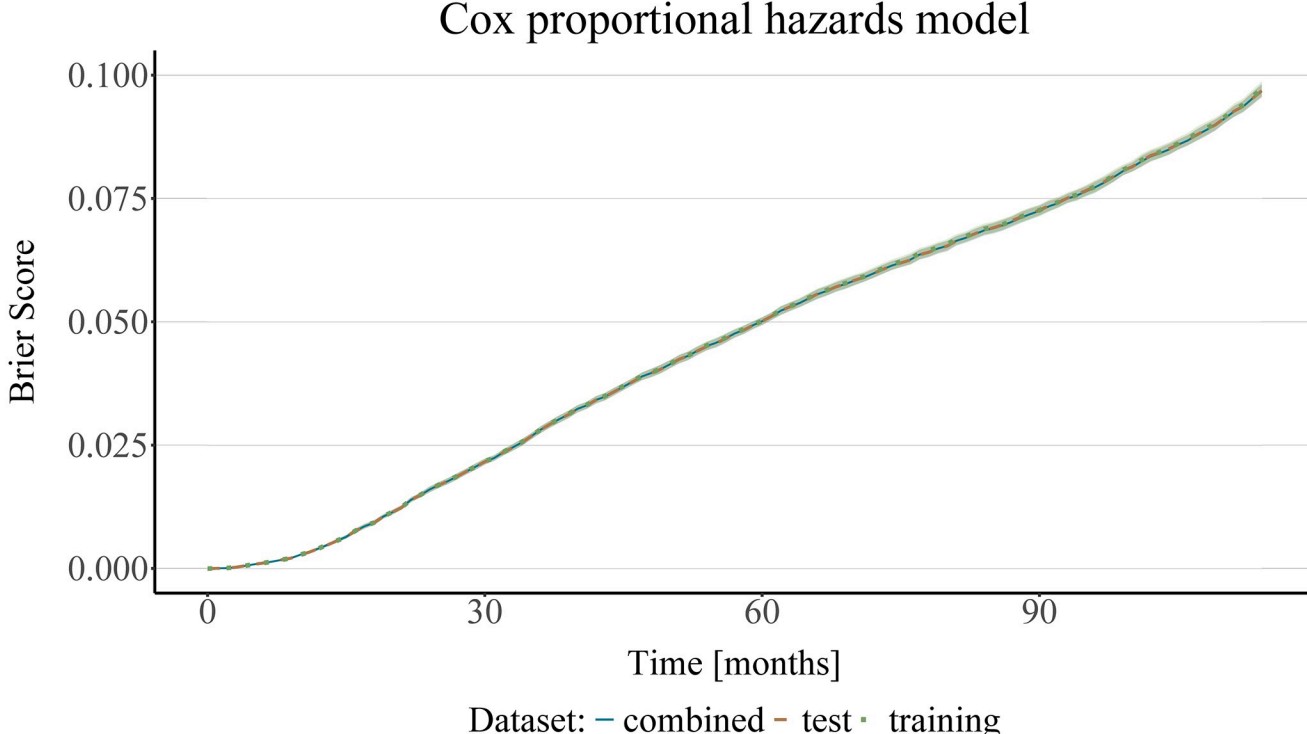

**Fig 6. Analysis of the SEER breast cancer data using a Cox proportional hazards model.** The plot shows the mean (bold center line) and standard deviation (shaded area) of the IPCW Brier score obtained on 10 bootstrap test samples, with IPC weights estimated from either the training, test, or the combined dataset. A Cox proportional hazards model was used for estimating the survival and the censoring survival functions.

uncensored Brier score may thus serve as a point of reference (ground truth), here, the true survival probabilities remain unknown. Calculating the WSE is therefore not possible. The assessment of possible differences in performance as a result of fitting the censoring function to either the training set, test set or both thus remains purely qualitative. However, it is still possible to make some interesting observations. Firstly, the predictive performance does not seem to be severely impacted by the choice of the data used to fit the censoring function. Using the training or the test set, or the combined dataset comprising both, leads to very similar results, see Fig 6.

Secondly, it is important to stress the importance of prior tuning of the ML models. In order to illustrate this, Figs 7 and 8 show the prediction error curves for both the untuned models (with the hyperparameters set to the default values pre-specified in the respective R packages, see S2 File), as well as for the models where the most important hyperparameters were tuned. As seen in Figs 7 and 8, we can observe differences in performance with respect to the data used for fitting the censoring survival function when the models are not properly tuned. This changes with prior tuning of the models, whereby we observe a convergence of the Brier score error curves. However, as we do not know the "ground truth", we cannot definitively say which dataset is best suited for model fitting. Interestingly, the differences between the predictive performance of the tuned and the untuned models are less pronounced for the XGBoost models than for the random forest models. This may be because the default settings of the XGBoost algorithm could have been closer to the "optimal" values than those of the random forest for this particular dataset. Additional figures showing only the mean values of the IPCW Brier Score for better legibility are available in S1 Fig.

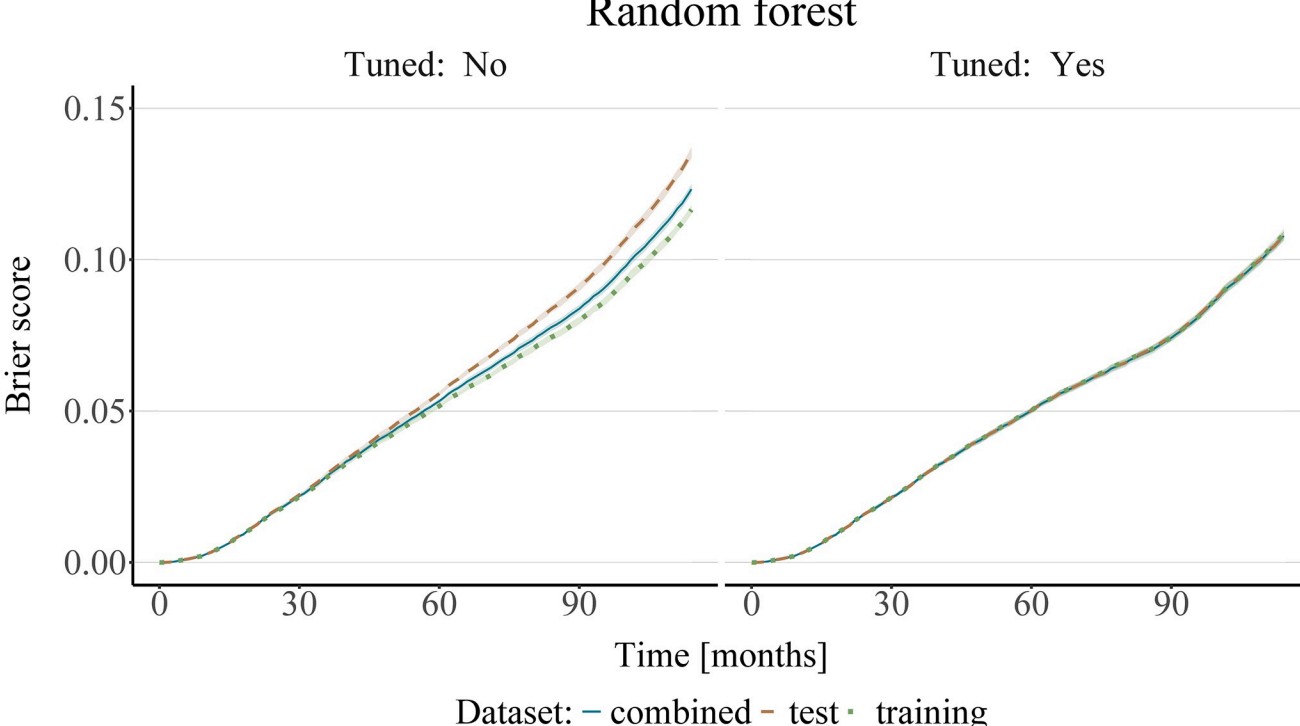

**Fig 7. Analysis of the SEER breast cancer data using random forest.** The plots show the mean (bold center line) and standard deviation (shaded area) of the IPCW Brier score obtained on 10 bootstrap test samples, with IPC weights estimated from either the training, test, or the combined dataset. A random forest was used for estimating the survival and censoring survival functions. The left panel shows the results of an untuned model, with the number of trees set to 500 and all other hyperparameters set to their default values, whereas the model in the right panel was tuned using Bayesian optimization.

In addition to analyzing the preprocessed SEER data described above, we repeated our study using a modified semi-synthetic version of the SEER data, following the methodology of Qi et al. [43]. Further details regarding the methodology and the results of the semi-synthetic analyses can be found in S3 File.

## Discussion

Inverse probability weighting has become a widely used approach in many statistical disciplines, including the analysis of data with missing values [44], propensity score adjustments for average treatment estimation in causal analyses [45, 46], the correction for selection bias in cohort studies [47], and the evaluation of predictive performance of mixture cure models in oncology [48]. In recent years, it has also become an integral part of causal ML [49, 50]. Here we considered the use of IPW in the analysis of right-censored time-to-event data. More specifically, we investigated the behavior of the IPCW Brier score to measure the predictive performance of time-to-event models. The Brier score is a popular performance criterion especially in comparison studies and benchmark experiments, being model-free in the sense that its definition does not rely on specific restrictions like the proportional hazards assumption or a parametric survival distribution. In these comparisons, IPC weights are commonly used to adjust the Brier score for a possible censoring bias that would occur from using the *observed* (and possibly censored) event times instead of the respective (partly unobserved) *true* event times.

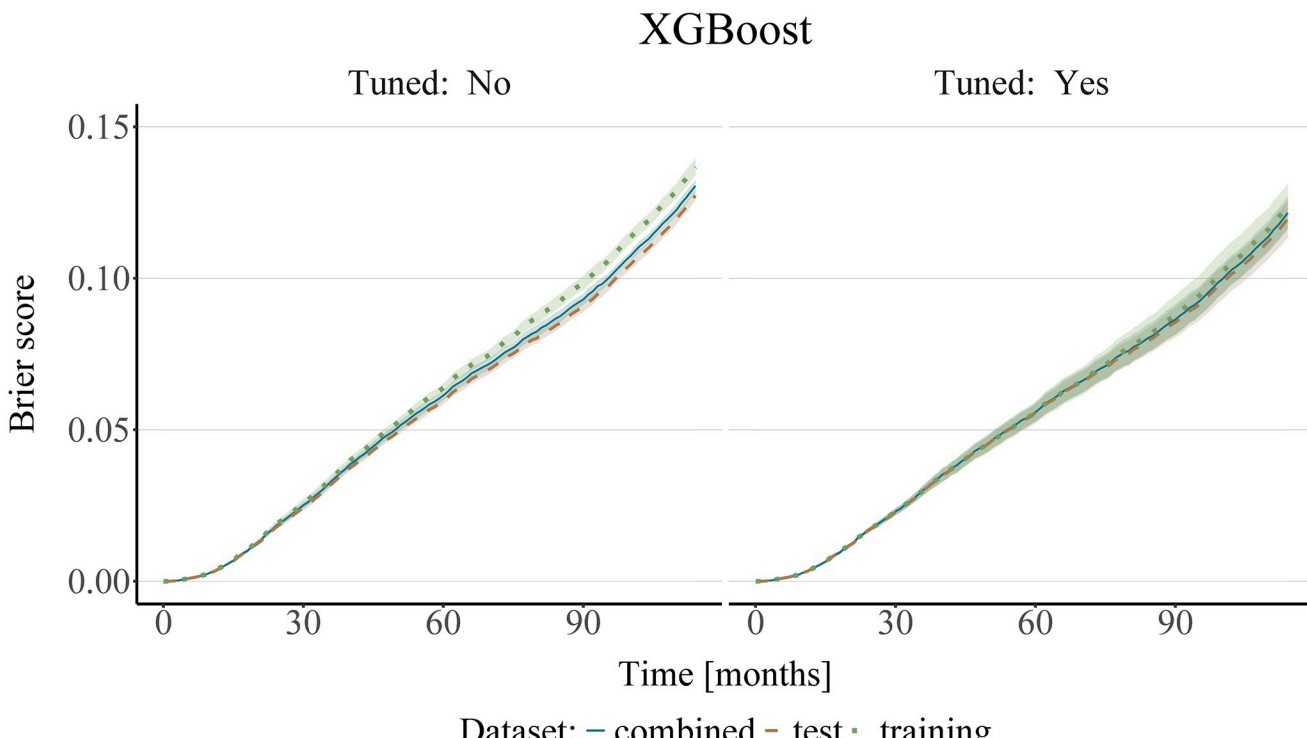

**Fig 8. Analysis of the SEER breast cancer data using XGBoost.** The plots show the mean (bold center line) and standard deviation (shaded area) of the IPCW Brier score obtained on 10 bootstrap test samples, with IPC weights estimated from either the training, test, or the combined dataset. XGBoost was used for estimating the survival and censoring survival functions. The left panel shows the results of an untuned model, with the number of boosting rounds set to 500 and all other hyperparameters set to their default values, whereas the model on the right was tuned using cross-validation and Bayesian optimization.

Our analysis focused on the question on how to best compute the IPC weights from a set of available data. More specifically, in the context of model development and validation studies investigating the predictive power of newly derived time-to-event models, we analyzed whether the IPC weights should be better computed from the training data, from the test data, or from the combined dataset, consisting of both the training and test data. We also analyzed the behavior of ML techniques when used to model the censoring process, investigating in particular the effects of "noisy" data and of proper vs. improper tuning of the hyperparameters on the estimated values of the IPC weights. Interestingly, though of high importance for data scientists, these issues have been largely neglected in the literature [21].

The main results of our numerical analysis are as follows:

(i). If one can safely assume that the training and the test data are subject to the same underlying censoring process, and when the latter is adequately represented by a covariate-free "marginal" censoring model, then it is largely irrelevant whether the training or the test data are used for the estimation of the IPC weights. In particular, our numerical experiments do not indicate any over-optimism due to the computation of the IPC weights from the test data. The only requirement is that the sample size and event rate of the dataset used for estimation should not be too small, ensuring a sufficiently stable fit of the censoring model. In cross-validation analyses, we therefore recommend to use the whole dataset to compute the IPC weights and to apply the same weights to all test folds when computing the IPCW Brier score.

(ii). Using a misspecified marginal censoring model when the true censoring process depends on the predictors will introduce a bias in the estimation of the IPCW Brier score. In these situations, it is important to incorporate the predictor variables in an adequately specified censoring model. Interestingly, even when the censoring model is specified correctly, our simulation study suggests that using the test data for IPC weight estimation generally results in slightly better estimates of the Brier score than using the training data or the combined dataset. We therefore recommend to use the test set for estimating IPC weights when the training and the test data are subject to the same covariate-dependent censoring process.

(iii). If a ML method is applied to estimate the IPC weights, it is of high importance to build a censoring model that is properly regularized. In particular, as demonstrated by our analysis of the SEER data, improper tuning of the model's hyperparameters (e.g., caused by applying some "rules of thumb" not adapted to the characteristics of the data) can lead to a substantial bias in the estimation of the Brier score. We therefore recommend to optimize the hyperparameters of the censoring model using nested cross-validation or resampling techniques.

(iv). Results from our high-noise scenario suggest that when dealing with noisy data, the censoring model should be fitted on either the test set or the combined dataset of training and test set rather than on the training set alone.

(v). If the censoring processes of the training and the test data are systematically different, then the test data should be used to compute the IPC weights. Since the calculation of the IPCW Brier score is rather resistant to a possible overfit of the censoring model (see (i) above), this strategy ensures that the IPC weights reflect the same censoring mechanism as the one applying to the test data's event times. Note that deviations in the censoring processes are often encountered when the Brier score is estimated from a set of external test data, the latter being often collected in settings that are systematically different from those of the model development phase.

A further important result of our simulation study is that the accuracy of the IPCW Brier score can be remarkably low in small-sample (or low-event-rate) scenarios. In fact, our numerical results show that it is not uncommon to encounter deviations as high as 0.2–0.3 between the IPCW Brier score and the expected uncensored Brier score (the latter representing the true prediction error, which is usually bounded between 0 and 0.25 in models predicting better than chance!). As a consequence, the predictive evaluation of time-to-event models may become highly questionable when sample sizes and event counts are "small".

Apart from the IPCW Brier score for right-censored time-to-event data, there are numerous other statistical measures and approaches that involve the computation of IPC weights. For example, the performance of time-to-event models is often evaluated using measures of discrimination such as the concordance index (C-index) for time-to-event data [51, 52]. Estimation of the C-index faces the same problems as the Brier score (including a possible bias when censored individuals are ignored, [53]), and it has become standard to use IPC weights for deriving censoring-adjusted estimates of the C-index [36]. As an alternative to the mean squared error, one could also consider IPCW-adjusted versions of the mean absolute error (MAE), as proposed by Qi et al. [43] and Schmid et al. [32]. By definition, the MAE penalizes extreme deviations between predicted and observed survival probabilities less strictly than the squared-error based IPCW Brier Score considered in this paper. A comprehensive discussion of performance metrics for time-to-event models (also including measures of calibration, margin-based metrics and estimators based on surrogate event times like pseudo-values) has been

provided by Qi et al. [43]. IPC weights are also used regularly in statistical modeling, e.g., for obtaining consistent estimates of subdistribution hazards in scenarios with competing events [54] and to fit ensembles of accelerated failure time models by least squares estimation [41]. Analogously, IPC weights can be applied to alternative censoring schemes than right censoring; for example, Tutz & Schmid [55] and Schmid et al. [56] considered IPC-weighted versions of the Brier score and the C-index, respectively for discrete time-to-event data. Although conceptual differences exist between the aforementioned uses of IPC weighting, we expect our findings to translate to these models and approaches as well.

## Supporting information

**S1 File. Brier score decomposition.**
(PDF)

**S2 File. Tuning details.**
(PDF)

**S3 File. SEER application: Semi-synthetic analysis.**
(PDF)

**S1 Table. Simulation: Additional results.**
(PDF)

**S1 Fig. SEER application: Additional figures.**
(PDF)

## Acknowledgments

We thank the academic editor and the reviewer for their valuable comments and suggestions. We also thank Alina Schenk and Moritz Berger for reviewing parts of the attached R code.

## Author Contributions

**Conceptualization:** Thomas Prince, Andrea Bommert, Jörg Rahnenführer, Matthias Schmid.

**Formal analysis:** Thomas Prince.

**Investigation:** Thomas Prince.

**Methodology:** Thomas Prince, Matthias Schmid.

**Software:** Thomas Prince.

**Supervision:** Matthias Schmid.

**Visualization:** Thomas Prince.

**Writing – original draft:** Thomas Prince.

**Writing – review & editing:** Andrea Bommert, Jörg Rahnenführer, Matthias Schmid.

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
