## [Decision Letter · Decision Letter 0]

8 Nov 2024

PONE-D-24-45479On the estimation of inverse-probability-of-censoring weights for the evaluation of survival prediction errorPLOS ONE

Dear Dr. Prince,

Thank you for submitting your manuscript to PLOS ONE. After careful consideration, we feel that it has merit but does not fully meet PLOS ONE’s publication criteria as it currently stands. Therefore, we invite you to submit a revised version of the manuscript that addresses the points raised during the review process.

We look forward to receiving your revised manuscript.

Kind regards,

Li-Pang Chen

Academic Editor

PLOS ONE

Additional Editor Comments:

Dear authors:

Thank you very much for submitting this manuscript to the journal. Our referee has reviewed your manuscript and she/he provides positive feedback for your work. I would like to invite you to prepare a revised manuscript and re-submit the revision to the journal.

In addition to the referee's comment, I also recall that the published paper "Chen, L.-P. (2020). Causal inference for left-truncated and right-censored data with covariates measurement error. Computational & Applied Mathematics, 39:126. DOI: 10.1007/s40314-020-01152-4." also adopts the inverse-probability-of-censoring weights for the average treatment effect estimation in causal inference. I would suggest the authors to cite this reference to enhance the comprehension of literature review.

Reviewers' comments:

Reviewer's Responses to Questions

**Comments to the Author**

1. Is the manuscript technically sound, and do the data support the conclusions?

Reviewer #1: Yes

2. Has the statistical analysis been performed appropriately and rigorously? 

Reviewer #1: Yes

3. Have the authors made all data underlying the findings in their manuscript fully available?

Reviewer #1: Yes

4. Is the manuscript presented in an intelligible fashion and written in standard English?

Reviewer #1: Yes

5. Review Comments to the Author

Reviewer #1: This paper addresses an interesting question in the estimation paradigm for the IPCW-based Brier score. A disagreement exists in the literature regarding the optimal approach for splitting the dataset to estimate the censoring distribution. This study makes a meaningful contribution by using various configurations of synthetic datasets alongside the SEER breast cancer dataset to empirically identify the best strategy for model evaluation.

The authors have thoughtfully investigated an overlooked issue in the community. Although the findings align with intuition and are not groundbreaking, the work is rigorous and offers significant value. The manuscript is well-organized, with clear motivation for the research questions, and the authors provide a solid overview of the Brier score and IPCW weighting methods. The experimental design is thorough, addressing a range of scenarios and adding to the study’s robustness.

While the paper is largely ready for publication, there are a few optional enhancements that could further strengthen the manuscript. However, these are good to have, not a must:

1. Currently, the synthetic experiments compare the IPCW Brier score with the expected Brier (Eq. 4). Adding an additional comparison with the mean squared error (Eq. 2) might provide a more comprehensive perspective.

2.The current synthetic datasets are designed by heuristics, which means they can hardly represent the distributions in the real-world. Qi et al. [1] proposed a way to generate realistic semi-synthetic survival datasets which contains real-world features, real-world event distribution, and close-to-real-world censor distribution. Implementing experiments with these semi-synthetic datasets would further enhance the study’s applicability.

3. For completeness, a brief overview of the proofs for (1) the decomposition of the Brier score into discrimination and calibration components, and (2) the expected Brier score’s decomposition into the MSE and variance terms, would be valuable additions. Such can be done in the supplementary.

4. The conclusion discusses the concordance index as an alternative evaluation metric. Expanding this section to include other metrics, such as the mean absolute or squared error between predicted and actual event times, as suggested in [1], would provide a more comprehensive evaluation framework.

[1] Qi et al., An Effective Meaningful Way to Evaluate Survival Models, ICML 2024

6. PLOS authors have the option to publish the peer review history of their article (what does this mean?). If published, this will include your full peer review and any attached files.

Reviewer #1: No

---

## [Author Response · Author response to Decision Letter 0]

7 Jan 2025

Dear Professor Chen,

we thank you and the reviewer for evaluating our manuscript, and we highly appreciate your positive feedback on our work. Following your recommendation, we have prepared a revision of the paper. 

In particular, we have (i) conducted an analysis of a semi-synthetic version of the SEER breast cancer data, (ii) added two proofs on the decomposition of the Brier score, and (iii) included the suggested reference by Chen (2020).

Further details can be found in the document "Response to Reviewers".

Yours sincerely,

Thomas Prince

---

## [Decision Letter · Decision Letter 1]

15 Jan 2025

On the estimation of inverse-probability-of-censoring weights for the evaluation of survival prediction error

PONE-D-24-45479R1

Dear Dr. Prince,

We’re pleased to inform you that your manuscript has been judged scientifically suitable for publication and will be formally accepted for publication once it meets all outstanding technical requirements.

Kind regards,

Li-Pang Chen

Academic Editor

PLOS ONE

Additional Editor Comments (optional):

Reviewers' comments:

Reviewer's Responses to Questions

**Comments to the Author**

1. If the authors have adequately addressed your comments raised in a previous round of review and you feel that this manuscript is now acceptable for publication, you may indicate that here to bypass the “Comments to the Author” section, enter your conflict of interest statement in the “Confidential to Editor” section, and submit your "Accept" recommendation.

Reviewer #1: All comments have been addressed

2. Is the manuscript technically sound, and do the data support the conclusions?

Reviewer #1: Yes

3. Has the statistical analysis been performed appropriately and rigorously? 

Reviewer #1: Yes

4. Have the authors made all data underlying the findings in their manuscript fully available?

Reviewer #1: No

5. Is the manuscript presented in an intelligible fashion and written in standard English?

Reviewer #1: Yes

6. Review Comments to the Author

Reviewer #1: Thank you for providing the revised manuscript and the detailed responses. I am satisfied with the revisions they have made and with the authors’ thoughtful incorporation of the suggested additional experiments, proofs, and clarifications. In my view, the manuscript is now well-prepared and addresses all key points raised during the review process. I have no further issue and recommend it for publication.

7. PLOS authors have the option to publish the peer review history of their article (what does this mean?). If published, this will include your full peer review and any attached files.

Reviewer #1: **Yes: **Shi-ang Qi

---

## [Editor Report · Acceptance letter]

23 Jan 2025

PONE-D-24-45479R1 

PLOS ONE

Dear Dr. Prince, 

I'm pleased to inform you that your manuscript has been deemed suitable for publication in PLOS ONE. Congratulations! Your manuscript is now being handed over to our production team.

Kind regards, 

on behalf of

Dr. Li-Pang Chen 

Academic Editor

PLOS ONE